# Salvage Radiotherapy for Relapsed Prostate Cancer after Radical Prostatectomy Is Associated with Normal Life Expectancy

**DOI:** 10.3390/cancers16030534

**Published:** 2024-01-26

**Authors:** Gunnar Lohm, Franz Knörnschild, Konrad Neumann, Volker Budach, Stefan Schwartz, Susen Burock, Dirk Böhmer

**Affiliations:** 1Department of Radiation Oncology, Johanniter-Hospital Genthin-Stendal, 39576 Stendal, Germany; 2Charité—Universitätsmedizin Berlin, Corporate Member of Freie Universität Berlin and Humboldt-Universität zu Berlin, 10117 Berlin, Germany; 3Department of Hematology, Oncology and Tumor Immunology (Campus Benjamin Franklin), Charité—Universitätsmedizin Berlin, Corporate Member of Freie Universität Berlin and Humboldt-Universität zu Berlin, 10117 Berlin, Germany; franz-ludwig.knoernschild@charite.de (F.K.); stefan.schwartz@charite.de (S.S.); 4Institute of Biometry and Clinical Epidemiology, Charité—Universitätsmedizin Berlin, 10117 Berlin, Germany; konrad.neumann@charite.de; 5Radiation Oncology Vosspalais, Private Clinic, Voss-St. 44, 10177 Berlin, Germany; volker.budach@t-online.de; 6German Cancer Research Center (DKFZ) and German Cancer Consortium (DKTK), 69120 Heidelberg, Germany; 7Clinical Trial Office (Campus Mitte), Charité—Universitätsmedizin Berlin, 10117 Berlin, Germany; susen.burock@charite.de; 8Department of Radiation Oncology (Campus Benjamin Franklin), Charité—Universitätsmedizin Berlin, 10117 Berlin, Germany; dirk.boehmer@charite.de

**Keywords:** biochemical relapse, overall survival, prostate cancer, salvage radiotherapy, PSA doubling time

## Abstract

**Simple Summary:**

Radiotherapy is a treatment option for prostate cancer patients who have increasing prostate-specific antigen values after prostatectomy. However, this treatment has never been compared directly in a single study with the course of untreated relapsed prostate cancer. We analyzed the course of prostate-specific antigen values before and after radiotherapy by calculating prostate-specific antigen doubling times. Prolongation of these values after radiotherapy indicated slower progression with this treatment. The survival analysis also suggests the advantages of salvage radiotherapy. In our analysis, when comparing our cohort with an age-matched cohort from life tables, it was observed that patients had a normal life expectancy after radiotherapy for recurrent prostate cancer following prostatectomy.

**Abstract:**

In patients with prostate cancer (PCa), salvage radiotherapy (SRT) for biochemical progression (BP) after radical prostatectomy (RP) improves PCa-specific survival. However, no prospective randomized trials have compared the effect of SRT with untreated patients. In this analysis of 151 patients who received SRT for post-RP BP, we compared their overall survival (OS) with virtual, age-matched controls (*n* = 151,000) retrieved from government life tables. We also investigated the risk factors associated with BP and OS and compared the prostate-specific antigen (PSA) doubling times (DTs) before and after SRT for patients with BP. The median follow-up was 9.3 years for BP and 17.4 years for OS. The risk factors significantly affecting BP were Gleason score (*p* < 0.001), pre-SRT PSA (*p* = 0.003), and negative surgical margins (*p* = 0.003). None of these risk factors were associated with OS. In 93 patients with BP after SRT, the median PSADT was significantly prolonged compared with pre-SRT values (3.7 vs. 8.3 months, *p* < 0.001). The OS did not differ between patients and controls (*p* = 0.112), and life expectancy was similar, likely due to the survival benefit of SRT. The prolonged PSADT after SRT further supports the beneficial role of SRT in this patient population. However, subsequent treatments were not systematically recorded, which may have affected the results.

## 1. Introduction

Radical prostatectomy (RP) is a potential curative option in patients with prostate cancer (PCa) and limited disease, but subsequent biochemical progression (BP) is frequently recorded, requiring salvage therapy [1,2]. In a large study evaluating 1746 patients, BP after RP was observed in 25% of patients within 15 years [3]. In another prospective randomized trial evaluating the addition of antiandrogen therapy to salvage radiotherapy (SRT) for patients with PCa and BP after RP, the addition of bicalutamide was associated with improved overall survival (OS) compared to the placebo [4]. Notably, this difference was observed beyond a median follow-up of one decade and illustrates the slow progression of PCa. In a report on the natural course of 1997 patients with progression following RP, a median metastasis-free survival of 8 years and a median time to death of 5 years was observed thereafter [5]. Thus, the data on tumor-specific survival or OS in patients with PCa and BP with median follow-up times shorter than 10 years are less likely to be meaningful. This has resulted in the frequent use of BP after SRT as a surrogate endpoint in various studies. However, the results of such analyses were recently questioned by the Intermediate Clinical Endpoints in Cancer of the Prostate Working Group, demonstrating the insufficiency of this surrogate endpoint in predicting the course of PCa [6]. In their meta-analysis of 15 primary radiation therapy-based trials, prostate-specific antigen (PSA)-defined relapse was identified as a weak surrogate for OS. In addition, another large retrospective study investigating the association between post-treatment BP and mortality recently reported a significant impact of BP after RP on PCa-specific survival only for patients with a Gleason score of >7 and PSA doubling time (PSADT) of <1 year. However, the authors concluded that BP, as currently defined, is not a reliable estimator of PCa death [7].

In the previous analyses of our single-center cohort, the median follow-up time was only 82 months [8]. Therefore, we updated the follow-up and survival data of our cohort to verify the previous findings and compare the survival probability with an age-matched control population.

## 2. Materials and Methods

We updated the data from our cohort of 151 patients with PCa without lymph node metastasis who received SRT for biochemical relapse after RP between 1997 and 2004 [8]. The details of the applied radiotherapeutic techniques have already been reported [9]. Briefly, a target dose of 66.6 Gy was administered in single fractions of 1.8 Gy to the prostate bed, including periprostatic surgical clips and the bladder neck. For locally advanced pT3 and pT4 tumors, we included the region of the former seminal vesicles in the radiation field.

BP before starting SRT was defined by any increase in PSA after RP. We defined BP after SRT as a PSA increase of at least 0.2 ng/mL above the post-SRT-start PSA nadir, with a further increase thereafter, regardless of whether lower PSA values were detected between these two PSA measurements. In patients without decreasing PSA values after the initiation of SRT, BP was defined as a PSA increase of at least 0.2 ng/mL above the pre-SRT PSA value with a subsequent increase. In addition, we assumed BP if the treating urologists initiated hormonal treatment before the above-mentioned criteria were met. The date of BP after the initiation of SRT was the date of the confirmatory second PSA measurement or the date of the last PSA measurement before starting hormonal treatment.

All statistical analyses were performed using IBM SPSS Statistics version 29 (IBM Corporation, Armonk, NY, USA). PSADTs were calculated using R Version 4.2.1 (R Core Team 2022. R: A language and environment for statistical computing. R Foundation for Statistical Computing, Vienna, Austria).

We calculated the PSADT before SRT using all PSA values from the post-RP PSA nadir until the start of SRT. For patients who experienced BP after SRT, we calculated the PSADT in a similar fashion using the nadir after the initiation of SRT until the detection of BP according to the above-mentioned criteria or until the documented initiation of hormonal treatment. To include PSA values below the lower limits of detection for the respective laboratories in the calculation, we replaced them with 0.005 ng/mL. This value was chosen because it was half the value of the lowest detection limit of the involved laboratories for all patients with BP. To calculate the PSADT for individual patients, we initially conducted a simple regression analysis for each patient using log(PSA) as the dependent variable and time as the independent variable. Subsequently, we determined the PSADT using the following formula: PSADT = log(2)/slope, where the slope is obtained from the previously mentioned regression analysis. In patients with a negative PSADT (i.e., PSA values showing a decreasing trend), we assigned 100 months in line with previous descriptions in the literature [10]. We then compared these calculated PSADT values before and after SRT-start using the Wilcoxon signed-rank test [11].

Eight parameters with a potential impact on the success of SRT regarding BP and OS in the literature [12,13,14] were chosen for the Cox regression analyses: pre-RP PSA, age at the start of SRT, pre-SRT PSA, time from RP until the start of SRT, PSADT before the start of SRT, tumor stage, margin status, and Gleason score. The continuous variables were dichotomized at their median. The categorial and ordinal variables were dichotomized (margin status, R0 vs. R1; tumor stage, ≤pT2c vs. ≥pT3a) or classified into three categories (Gleason score ≤ 6 vs. 7 vs. ≥8). We also integrated the hormonal treatment initiated for BP subsequent to SRT into the Cox regression analysis of OS.

We retrieved the updated survival data for our patients from Charité patient charts and the Berlin registry of residents. We then compared these data to a virtual, age-matched cohort of controls from the normal population. Virtual controls were generated using life tables from the Federal Statistical Office of Germany [15]. Each patient was matched to 1000 virtual controls of the same age. We used Kaplan–Meier analysis and the log-rank test to compare our patient cohort with controls representing the normal male population [16]. For comparisons with the literature data, we created a table of patient characteristics for our cohort and patient characteristics from three larger, previously published studies [4,14,17]. In particular, we performed Cox regression analysis of OS, with pre-SRT PSA dichotomized at 0.7 ng/mL and pre-SRT PSADT calculated only with PSA values above the limit of detection, for comparison with previously published data [4,8,18].

## 3. Results

After a median follow-up of 9.3 years with respect to the biochemical control, we recorded BP in 93 patients (Figure 1). 

Seventy-two patients were known to have received hormonal ablative therapy subsequent to SRT-start due to BP. Cox regression analysis identified three variables with a significant impact on BP-free survival (BPFS): higher Gleason score (*p* < 0.001), higher pre-SRT PSA (*p* = 0.003), and negative margin status (*p* = 0.003; Table 1). 

The comparison of PSADT before and after SRT for patients with BP after the start of SRT revealed a significantly longer PSADT after SRT (median 3.9 months before vs. 8.3 months after SRT, *p* < 0.001; Figure 2).

The median follow-up for OS was 17.4 years. Cox regression analysis of the eight variables to assess their association with OS did not yield any significant results for any of the variables, whereas the initiation of hormonal treatment post-SRT revealed a statistically significant impact on OS (Table 2).

The log-rank test applied to Kaplan–Meier analyses revealed no significant difference in survival between our patient cohort and the virtual controls generated from life tables (*p* = 0.112; Figure 3).

Comparisons of the patient characteristics of our cohort with data from the literature revealed a lower median pre-SRT PSA value and a higher proportion of lower tumor stages at RP (i.e., pT2; Table 3).

In the additional analyses considering pre-SRT PSA levels dichotomized at 0.7 ng/mL and pre-SRT PSADT calculated using detectable PSA values exclusively, we observed that pre-SRT PSA had no significant effect on OS (*p* = 0.96, hazard ratio [HR] 1.016, 95% CI: 0.546–1.890). However, a notably significant effect on OS was found with a shorter PSADT (*p* = 0.013, HR 0.398, 95% CI 0.192–0.826).

## 4. Discussion

The evaluation of the potential beneficial effects of SRT in patients with BP after RP is challenging and requires long-term data, ideally with follow-up times beyond 10 years.

Several studies have explored the impact of SRT on BP after RP. In a review of 13 contemporary retrospective studies evaluating SRT, 12 of the studies provided data on biochemical relapse-free survival with median follow-up times ranging from 3 to 6.8 years [19]. However, the Kaplan–Meier curve for BPFS in our cohort revealed a disappointing finding of approximately two-thirds of the patients experiencing BP within 10 years. A similar Kaplan–Meier curve with only a 32% BP-free probability at 6 years was reported in a multi-institutional study with 1540 patients [14]. In contrast to these high rates of biochemical relapses, in a prospective randomized trial testing the concept of early SRT against adjuvant radiotherapy, the biochemical relapse rate was only 12% for the SRT group after a median follow-up of 5 years [20]. One reason for this lower rate of relapses may have been the higher PSA (0.4 ng/mL) cutoff for defining BP after SRT. Furthermore, no details about PSADT were reported, an important risk factor, as discussed below. Therefore, a comparison of these study data with our current results does not seem meaningful.

However, a comparison of our data with previously published studies confirmed that the rate of biochemical relapse continuously increases with longer follow-up times [8,9]. This raises questions regarding the efficacy and value of SRT as a therapeutic strategy. However, BPFS, which is frequently used as a relevant study endpoint, may not serve as a sufficient prognosticator in predicting the course of relapsed PCa after RP [6]. In a retrospective study of 501 patients, SRT was associated with a decreased risk of all-cause mortality after a median follow-up of 11.3 years [20]. In addition, a large matched-pair analysis comparing SRT and no SRT confirmed a positive impact of SRT on OS [17]. Our comparison of OS with survival data from an age-matched virtual sample of the general German male population revealed no significant difference, indicating not only improved OS but also normal life expectancy.

The PSADT is an important variable indicating the aggressiveness of PCa. In the aforementioned study on the natural course of the disease in patients with relapsed PCa after RP, a pre-RP PSADT of <10 months had a negative impact on metastasis-free survival [5]. The pre-SRT PSADT in our patient cohort was comparably short, with a median of only 3.93 months. Whether the prolonged PSADT after SRT in our study had an impact on OS remains unclear. In the retrospective study of 501 patients receiving SRT [20], an improved OS was observed, independent of the pre-SRT PSADT dichotomized at 6 months. However, this study did not explore the impact of SRT on the PSADT.

A comparison of the patient characteristics in our cohort and three larger studies [4,14,17] revealed a higher proportion of lower Gleason scores and lower median pre-SRT PSA values in our study. These more favorable parameters in our cohort may have a positive impact on the observed OS. However, compared to [14] that provided corresponding data, the median pre-SRT PSADT in our cohort was shorter, indicating greater aggressiveness of PCa among our patients.

Three modifications to SRT have been tested in an effort to improve outcomes with biochemical relapse after RP: the addition of systemic therapy, an escalation in the SRT dose, and an extension of the irradiated volume encompassing the pelvic lymphatics.

The addition of hormonal treatment improved OS among patients exhibiting pre-SRT PSA values ≥0.7 ng/mL [4] and has been implemented into current German guidelines for the treatment of PCa [21]. However, our analysis of pre-SRT dichotomized at 0.7 ng/mL could not detect a significant impact of this parameter on OS. Another prospective randomized trial, with the short-term application of goserelin added to SRT, showed similar results for BPFS and metastasis-free survival [22]. As the authors noted, their follow-up of ~10 years was too short to yield significant results for subgroup and OS analyses.

Two large prospective randomized phase III trials investigating a dose escalation of SRT have been published so far [23,24] with diverging results. One showed a benefit of SRT doses >70 Gy but only in patients with higher Gleason scores (i.e., 8–10). Both trials yielded sufficient data only for BPFS because of the median follow-up times of only 5 to 6 years.

Retrospective studies investigating whole pelvic radiotherapy (WPRT) compared with standard SRT to the prostate bed with or without added hormonal treatment revealed an improved BPFS in patients with different high-risk parameters [25,26,27]. These findings have been explored further in recently published prospective randomized trials. In a three-arm trial testing standard SRT (arm 1) versus SRT plus short-term androgen deprivation therapy (arm 2) versus the addition of pelvic radiotherapy to the second arm (arm 3), BPFS was superior in the third arm [28]. The authors concluded that a longer follow-up was needed to better define the influence of WPRT on distant metastasis and survival endpoints. An updated phase II study that included only high-risk and lymph node-positive patients with PCa showed comparable OS results for WPRT after a median follow-up of 10 years [29].

With PCa-directed radiotherapy based on positron emission tomography, a novel treatment strategy is emerging that goes beyond the modification of SRT. Refinements to this treatment strategy concerning radiation fields and the addition of hormonal treatments are unresolved [30], and the impact on survival endpoints has not been sufficiently investigated.

We performed a modified calculation of PSADT before and after SRT that accounted for PSA values below the detection limit. The comparison with our previous definition of PSADT showed a significant impact of pre-SRT PSADT on OS when excluding PSA values below the detection limits. This confirms that integrating PSA values below the detection limits into PSADT calculations describes the development of BP more precisely.

The median follow-up time in our analysis was beyond 17 years, and patients received homogeneous radiotherapy at a single center. However, our study is limited due to its retrospective nature and limited sample size. Furthermore, we were unable to provide details about further treatments, except for 72 individuals who were known to have received antihormonal treatment after they experienced BP after SRT. Any androgen deprivation therapies were not initiated according to a standardized protocol but at the discretion of the treating physicians. The reasons for the observed impact of these hormonal treatments on OS remain unclear and require further studies. Our analysis is based on incomplete data concerning treatments subsequent to SRT because information about the initiation of further treatments was not systematically captured, and data from 79 patients in our cohort were unavailable. However, our finding is in line with the results of the aforementioned randomized trial [4].

It remains unclear whether any modified SRT for relapsed PCa in patients without lymph node metastasis is superior to standard SRT as analyzed here, except for the addition of hormonal treatment for pre-SRT PSA values >0.7 ng/mL. Otherwise, improved OS beyond normal life expectancy is hardly conceivable, regardless of any salvage therapy modifications. Thus, standard SRT with the application of a radiation dose of ~66 Gy to the prostate bed could still be regarded as the standard of care for relapsed PCa after RP. Further treatment should be considered only for patients who experience a further PCa relapse, conducted in analogy to the concept of early SRT after RP compared with adjuvant radiotherapy to protect patients against the side effects from potentially unnecessary treatments. However, our study has limitations, and further confirmation of the OS findings is needed, especially from randomized trials with an update to follow-up times well beyond 10 years.

## 5. Conclusions

This study presents long-term survival data for 151 patients with PCa and BP after RP who were subsequently treated with SRT. The median observation time in these patients was 17.4 years, which allowed a meaningful comparison of their life expectancy with that of normal controls. We observed no difference in survival, which may be due to the benefit of SRT and the genuinely slow progression of PCa. We used a modified calculation of PSADT, which likely detects the biochemical evolution of PCa in the setting of salvage treatment more accurately. Our data support the positive impact of SRT in patients with relapsed PCa after RP, and this impact was associated with a normal life expectancy.

## Figures and Tables

**Figure 1 cancers-16-00534-f001:**
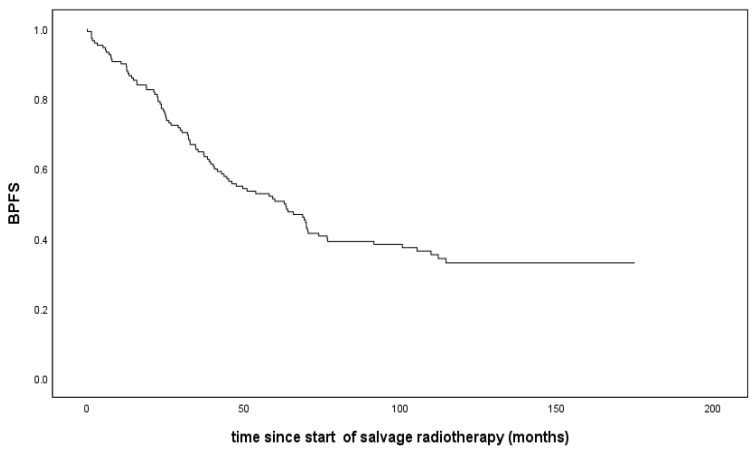
Biochemical progression-free survival (BPFS) in 151 patients who underwent salvage radiotherapy after radical prostatectomy.

**Figure 2 cancers-16-00534-f002:**
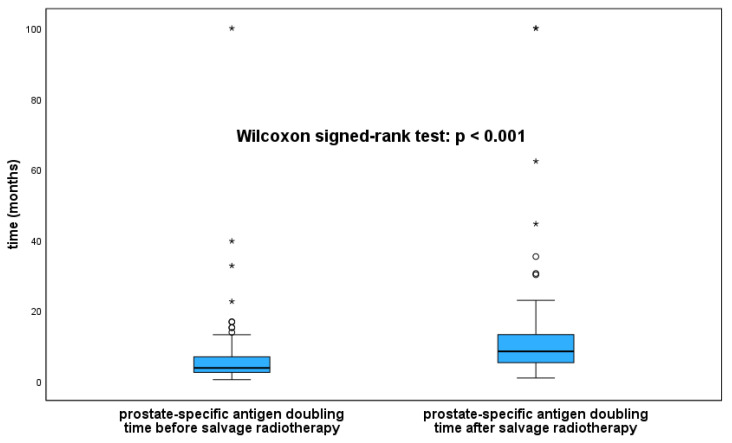
Boxplots of prostate-specific antigen doubling times before and after salvage radiotherapy in 93 patients with biochemical relapse after radiotherapy. Box = interquartile range; bold line = median; whiskers = range from minimum to lower quartile and upper quartile to maximum, respectively; rings = outliers ≥ 1.5 × interquartile range above upper quartile; asterisks = extreme values > 3 × interquartile range above upper quartile.

**Figure 3 cancers-16-00534-f003:**
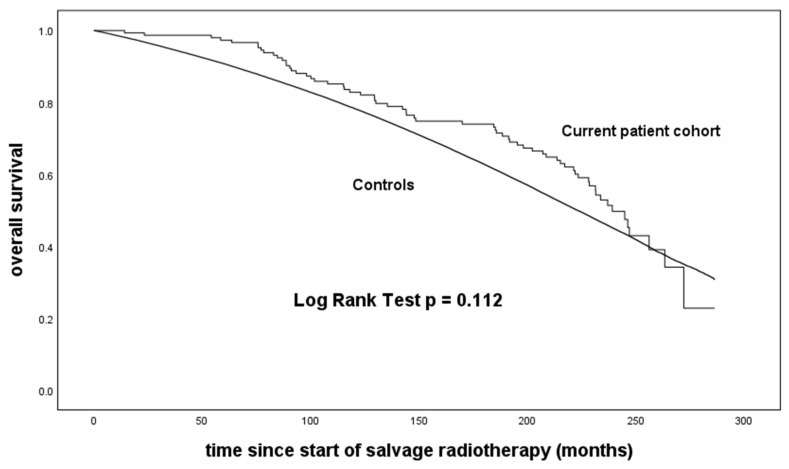
Kaplan–Meier curves of overall survival in our patient cohort compared with virtual matched controls (*n* = 151,000) generated from life tables.

**Table 1 cancers-16-00534-t001:** Cox regression analysis of biochemical progression-free survival over a median biochemical follow-up of 9.3 years.

Parameter	Median (Range) *	*p*-Value, HR (95% Confidence Interval)
age at SRT-start (years)	64.6 (53.2–80.9)	0.98 HR 0.993 (0.614–1.605)
pre-RP PSA (ng/mL)	12.0 (2.0–106.8)	0.93 HR 0.980 (0.615–1.561)
time from RP to start of SRT (months)	19.1 (1.58–165.8)	0.14 HR 0.663 (0.384–1.143)
pre-SRT PSA (ng/mL)	0.344 (0.034–8.871)	**0.003** HR 2.151 (1.307–3.539)
pre-SRT PSADT (months)	3.9 (0.258–100)	0.06 HR 0.602 (0.357–1.014)
tumor stage	≤T2c vs. ≥T3a	0.08 HR 1.555 (0.950–2.537)
margin status	R0 vs. R1	**0.003** HR 0.455 (0.272–0.762)
Gleason score	≤6 vs. 7 vs. ≥8	**<0.001** HR 3.207 (1.649–6.237)

* If not stated otherwise. HR = hazard ratio. Significant values are in bold.

**Table 2 cancers-16-00534-t002:** Cox regression analysis of overall survival over a median follow-up of 17.4 years.

Parameter	Median (Range) *	*p*-Value, HR (95% Confidence Interval)
age at SRT-start (years)	64.6 (53.2–80.9)	0.12 HR 1.563 (0.888–2.750)
pre-RP PSA (ng/mL)	12.00 (2.0–106.8)	0.98 HR 0.991 (0.570–1.723)
time from RP to SRT-start (months)	19.1 (1.58–165.8)	0.80 HR 1.087 (0.565–2.091)
pre-SRT PSA (ng/mL)	0.344 (0.034–8.871)	0.79 HR 1.091 (0.583–2.041)
pre-SRT PSADT (months)	3.9 (0.258–100)	0.61 HR 1.184 (0.616–2.276)
tumor stage	≤T2c vs. ≥T3a	0.09 HR 1.728 (0.911–3.278)
margin status	R0 vs. R1	0.052 HR 1.949 (0.995–3.817)
Gleason score	≤6 vs. 7 vs. ≥8	0.94 HR 0.968 (0.433–2.167)
HT for BP after SRT	known (*n* = 72) vs. not known (*n* = 79)	**0.01** HR 2.388 (1.219–4.679)

* If not stated otherwise. HR = hazard ratio. HT = hormonal treatment. BP = biochemical progression. Significant value in bold.

**Table 3 cancers-16-00534-t003:** Comparison of patient characteristics in our study with three other patient cohorts from the literature.

	Current Study (*n* = 151)	Shipley [4] (*n* = 760)	Stephenson [14] (*n* = 1540)	Tilki [17] (*n* = 437)
median age at prostatectomy (years)	62.1	n. r.	62	63.8 (mean)
median age at SRT-start (years)	64.6	65	n. r.	n. r.
median pre-RP PSA	12.0	n. r.	10.5	12.1 (mean)
median time from RP to SRT-start (months)	19.1	25.2	15	n. r.
median pre-SRT PSA	0.34	0.6	1.1	n. r.
median pre-SRT PSADT (months)	3.9	n. r.	6.9	n. r.
median follow-up (years)	9.3	13 (surviving patients)	4.5	8
tumor stage: pT2	48.4%	32.6%	n. r.	51.9%
pT3a	30.5%	n. r.	65%	31.1%
pT3b	16.6%	n. r.	24%	15.6%
∑ pT3a + pT3b	47.1%	66.7%	89%	46.7%
margin status R0	37.7%	25.1%	49%	61.8%
R1	55.6%	74.9%	51%	38.2%
Gleason score ≤ 6	44.3%	28.2%	26%	30%
7	35.8%	54.5%	52%	57.7%
≥8	18.5%	17.3%	22%	12.3%

n. r.: not reported.

## Data Availability

Research data are stored in an institutional repository and will be shared upon request to the corresponding author.

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
