# Peer review of "Salvage Radiotherapy for Relapsed Prostate Cancer after Radical Prostatectomy Is Associated with Normal Life Expectancy"

_cancers, 2024, doi:10.3390/cancers16030534_

Round 1

Reviewer 1 Report

Comments and Suggestions for Authors

This study presented long-term survival data for 151 patients with PCa and BP who subsequently underwent SRT. The present study showed that the life expectancy of PCa patients who underwent SRT for BP was similar to that of normal individuals. I would like to point out several points of improvement in this research report.

Minor

Seventy-two patients had received HT prior to starting SRT, and the impact of HT needs to be examined and discussed.

1.In what cases did you use HT in combination?

2.Could the combination of SRT and HT have affected the OS? Please consider including SRT only vs SRT+HT in the parameters of Table 2.

3.Please consider including the difference between SRT only and SRT+HT in the Discussion portion.

Other minor point

4. The right side of Figure 2 is labeled prostata-specific antigen, correct the spelling.

Author Response

1.-3. : This is a misunderstanding: None of our patients received HT prior to or during SRT. HT was initiated  by the treating urologists for treatment of further biochemical progression after SRT. The potential impact of further treatment, known or not (only known for 72 patients) will be mooted in the expanded discussion section we are working on.

4. The wrong spelling will be corrected.

Reviewer 2 Report

Comments and Suggestions for Authors

Dear authors,

Congratulations on your hard work. Here are my recommendations:

1. Recording and analyzing subsequent therapies after SRT could offer a more holistic view of disease management and its impact on long-term survival.

2. Detailing the reasons behind initiating hormonal treatment could provide context and enhance the study's robustness.

3. Acknowledging limitations, such as the retrospective nature of the study and the absence of detailed subsequent treatments, would strengthen the conclusions and set realistic expectations.

Author Response

1.: The comment is absolutely correct. Unfortunate we are not able to provide sufficient data on further treatments after SRT, as mentioned in the last paragraph of the discussion section. We will further discuss this item in the expanded discussion section we are working on.

2.: HT was initiated at the discretion of the treating urologists. Unfortunately, we do not know motives and details of the decisions of the involved colleagues for initiating HT in the vast majority of cases.

3.: The comment is correct. However, in the last paragraph of the discussion section we have already done so.

Round 2

Reviewer 2 Report

Comments and Suggestions for Authors

In the current form I consider the manuscript acceptable for publication.